# Modeling Integrated Impacts of Climate Change and Grazing on Mongolia's Rangelands

**Virginia Anne Kowal** [1,*] [iD], **Julian Ahlborn** [2] [iD], **Chantsallkham Jamsranjav** [3], **Otgonsuren Avirmed** [3] and **Rebecca Chaplin-Kramer** [1]

1   Natural Capital Project, Woods Institute for the Environment, Stanford University, Stanford, CA 94305, USA; bchaplin@stanford.edu
2   Sustainable Grassland Systems, Leibniz Centre for Agricultural Landscape Research, D-14641 Paulinenaue, Germany; julian.ahlborn@zalf.de
3   Wildlife Conservation Society Mongolia, Ulaanbaatar-14200, Mongolia; jchantsallkham@wcs.org (C.J.); oavirmed@wcs.org (O.A.)
*   Correspondence: gkowal@stanford.edu

**Abstract:** Mongolia contains some of the largest intact grasslands in the world, but is vulnerable to future changes in climate and continued increases in the number of domestic livestock. As these are two major drivers of change, it is important to understand interactions between the impact of climate and grazing on productivity of Mongolia's rangelands and the livelihoods they sustain. We use a gridded, spatially explicit model, the Rangeland Production Model (RPM), to explore the simultaneous and interacting effects of climate and management changes on Mongolia's rangeland and future livestock production. Comparing the relative impact of temperature, precipitation, and grazing intensity, varied individually and in combination, we find that climatic factors dominate impacts on forage biomass and animal diet sufficiency. Site rainfall strongly mediates the impact of grazing on standing biomass, such that more productive or higher-rainfall sites are more vulnerable to increases in grazing pressure. Gridded simulations covering Mongolia's Gobi-Steppe ecoregion show that while rangeland biomass is generally predicted to increase under future climate conditions, interactions among spatially varying drivers create strong heterogeneity in the magnitude of change.

**Keywords:** grazing impact; climate change; rangeland condition; ecological model

## 1. Introduction

Mongolian rangelands are a globally significant resource, as of one of the largest remaining intact rangeland ecosystems on earth. They provide important habitat for native wildlife, including threatened and endangered species such as the golden eagle (*Auquila schysaetos*), snow leopard (*Panthera uncia*) and the wild ass or Khulan (*Equus hemionus hemionus*) [1]. These rangelands have also supported traditional nomadic pastoralism for over 1000 years, with a rich cultural history and encompassing vibrant traditional ecological knowledge. With mobile, flexible extensive herding, about one third of the total population in Mongolia directly benefit from rangeland resources [2].

Mongolia's rangelands have also undergone rapid change and face persistent threats. The country's transition from a collectivist to a market economy and the resulting privatization of livestock after 1990 brought significant changes to nomadic pastoralism, including a quadrupling of livestock densities since 1970 [3]. Increasing global demand for cashmere has driven particularly dramatic increases in the number of goats held by herders in Mongolia; as one of the world's largest producers of cashmere, the number of domestic livestock in Mongolia continues to grow [4].

Along with drastic changes in livestock numbers, Mongolia is experiencing large shifts in rangeland productivity due to climate change. Over the past half century, steadily rising temperatures and shifting rainfall patterns have already driven large changes in

forage biomass [5]. Dryland ecosystems such as Mongolia are especially sensitive to climate change, where even small changes in temperature and precipitation can result in a major ecosystem response [6–8]. The Fifth Assessment Report (AR5) of the Intergovernmental Panel on Climate Change (IPCC) projected large increases in temperature and annual precipitation for Mongolia under all future emission scenarios; temperatures could increase by 6 °C and precipitation by 30% by the mid-21st century, according to ensemble-mean projections under the high-emission RCP8.5 scenario [9].

Several previous studies have used ecosystem modeling to explore potential consequences of the changing climate for Mongolia. An analysis using the Century ecosystem model, with climate projections from the AR5, suggested that changes in aboveground biomass would be highly variable between regions of the country [10]. Based on general trends evident in the IPCC Fourth Assessment Report (AR4), Angerer et al. [11] drew on a heuristic understanding of plant growth to predict that net primary productivity would decline in most of the important rangeland areas of Mongolia. A recent global modeling study, using the G-range global rangeland model and climate projections from the AR5, also suggested that rangeland productivity in Mongolia would decrease by the year 2050, partly due to modeled changes in forage community composition [12]. Future climate projections at the regional scale, and their expected ecosystem consequences, have varied widely with advances in global climate models and with updates to future emission scenarios. Our study is the first, to our knowledge, to analyze the latest climate projections for Mongolia from the recently completed Coupled Model Inter-comparison Project Phase 6 (CMIP6), which will form the basis of the Sixth Assessment Report (AR6) [13].

While existing studies have addressed the potential impacts of climate change on Mongolia's rangelands, few have included any analysis of simultaneous changes in grazing pressure. Understanding the combined impacts of these two important drivers on rangeland condition is critical, and yet is rarely addressed [14]. Both long-term climatic change and overgrazing by livestock have been pinpointed as drivers of degradation in semi-arid rangelands [15,16], and these two factors interact. In semi-arid and arid rangeland systems, interannual variability in rainfall is a determining factor both of forage production [17] and of the response of the rangeland ecosystem to grazing [18]. Observations of this interaction between climatic variability and grazing gave rise to the concept of non-equilibrium dynamics in rangelands [19,20]. While the conceptual boundaries between equilibrium and non-equilibrium systems have been a topic of debate for decades [21–23], Mongolia's strong climatic gradients and long history of nomadic grazing have made it an especially valuable setting in which to explore the validity of the non-equilibrium concept [24,25].

A key aspect of non-equilibrium dynamics in Mongolia's rangelands is a natural phenomenon called dzud. A dzud is an extreme mortality event, where up to 40% of livestock may perish in a single winter [3]. The time between the dieback of livestock and the recuperation of the herds acts as a resting period for the vegetation, which is an important factor contributing to the resilience of these ecosystems. While some dzud years show an obvious spike in mortality at the national level, many dzud are more localized events that reflect spatial heterogeneity in climatic conditions, animal density, and preparedness by herders [26,27]. As with dzud, future forage availability is expected to be heterogeneous in space. A retrospective analysis of trends in aboveground biomass in Mongolia's grasslands in the recent past showed that spatial heterogeneity of these trends has been extreme: some areas saw increases of up to 40% in annual peak biomass, while other areas exhibited decreases of up to 65% [5]. With expectations that future changes will exhibit similar or elevated spatial and temporal variability, understanding climate–grazing interactions in a framework that accounts for this variability is of critical importance.

Here, we introduce and apply an updated, gridded version of the Rangeland Production Model (RPM) to fill key gaps in understanding of the future of Mongolia's rangelands. We analyze, for the first time to our knowledge, the latest climate projections for Mongolia from the CMIP6 global climate modeling project [13]. Our study is also the first to address simultaneous changes in climate and animal management for Mongolia in the future,

thereby exposing important interactions between these two major drivers. We use RPM to contrast separate and combined impacts of changes in climate and livestock density on rangeland and animal condition, running fixed-level contrasts at 15 sites with validation data, and with spatially explicit climate scenarios across a large portion of Mongolia. With this set of simulations, we address the following questions: (1) What are the relative impacts of reasonable future changes in temperature, precipitation, and animal density on rangeland and animal condition? (2) How do these impacts vary across a rainfall gradient? (3) How does predicted future rangeland and animal condition vary spatially?

## 2. Materials and Methods

### 2.1. Model Description

We use the Rangeland Production Model (RPM; v.0.2.0) to quantify spatially varying impacts of climate and livestock on rangeland and animal condition. RPM is a gridded ecosystem model for extensive ruminant grazing systems, composed of a gridded (i.e., pixel- or raster-based) implementation of the Century ecosystem model [28] coupled with a basic animal physiology sub-model adapted from GRAZPLAN [29]. The plant production sub-model, which replicates the herbaceous biomass production routine of Century, simulates the growth of herbaceous forage on each pixel of the simulated area, according to climate and soil inputs supplied by the user, at a monthly timestep. The ruminant physiology sub-model adapted from GRAZPLAN calculates the offtake of forage by grazing animals according to the biomass and protein content of the simulated forage and estimates the adequacy of the diet to meet the animals' energy requirements using a simplified metric, diet sufficiency. Then, the estimated offtake by animals is integrated into the regrowth of forage in the following timestep through impacts on simulated potential production, root:shoot ratio, and plant nitrogen content, according to Century's existing grazing routine [30]. The spatial extent and resolution of RPM are determined by model inputs; here, gridded simulations were run at a resolution of 10 arc minutes. Input data sources for this application are described briefly in Section 2.3, and in full in the Supplementary Materials. An earlier, point-based version of the model was described by [31]. A full model description of RPM v0.2.0 is provided in the Supplementary Materials.

### 2.2. Study Area

Mongolia is a landlocked country surrounded by Russia and China, located at 41.5–52° N latitude and 87.5–119.5° E longitude (Figure 1). It is surrounded by high mountains and is a country of predominantly high elevation: the average altitude is 1580 m above sea level. The country is characterized by a harsh continental climate due to its central Eurasian location and high elevation, with short, dry summers and long, cold winters. Precipitation is relatively low across the country and shows a strong latitudinal gradient from north to south. Driven by the strong precipitation and altitudinal gradients, vegetation is dominated by herbaceous forage over much of the country, transitioning to mountain taiga (forest) and alpine vegetation in the north and west. For a summary of current climatic conditions at the study locations shown in Figure 1, see Appendix A, Table A1.

### 2.3. Sensitivity of Rangeland and Animal Condition to Temperature, Precipitation, and Animal Density

We contrasted the impacts of changing temperature, precipitation, and animal density on rangeland and animal condition at 15 sites arrayed along a 600 km precipitation gradient across Mongolia (Figure 1; Appendix A, Table A1), where biomass and grazing intensity data had previously been collected [33]. At each site, we ran RPM for a circular area of approximately 24 km$^2$. We summarized model outputs for each site as the mean value across pixels within the circular area.

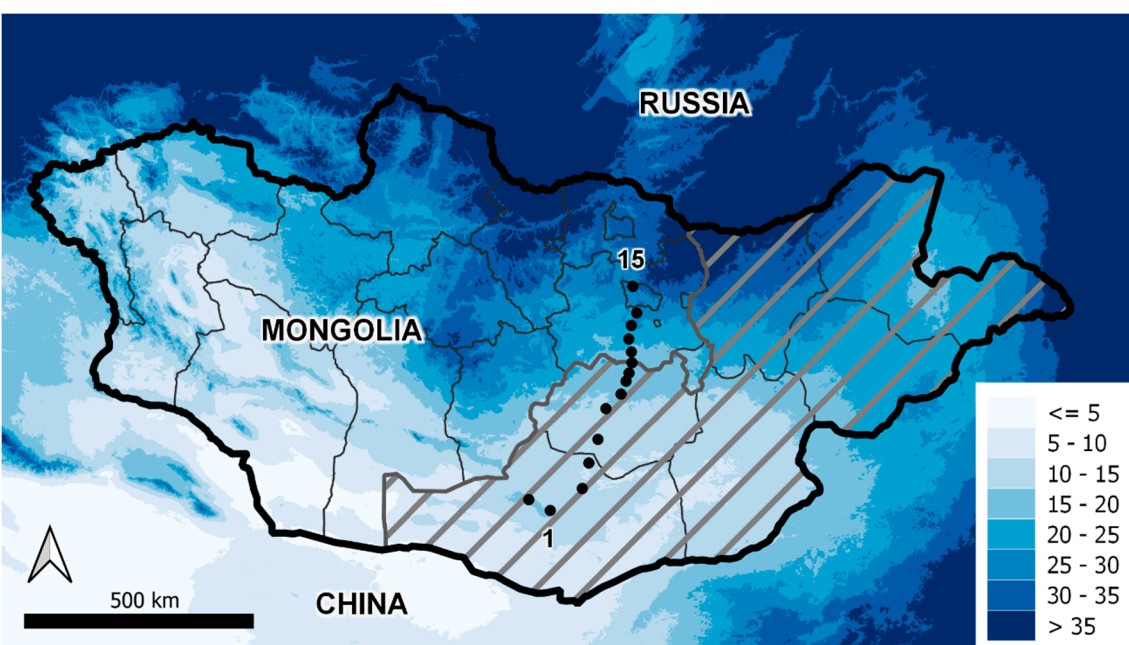

**Figure 1.** Location and average annual precipitation (cm; [32]) at simulated sites in Mongolia. Fixed contrasts were applied at 15 sites (black points) arrayed across a precipitation gradient in the center of the country. The hashed area shows the Gobi-Steppe ecoregion, where gridded simulations were conducted. Sites were numbered 1–15 from south to north; for additional site information, see Appendix A, Table A1.

We conducted simulations at these 15 sites because they offer strategic placement along a precipitation gradient, and because existing field data were available there to inform model calibration and validation. The calibration and validation results are described briefly at the end of this section, and in full in Appendix C. The field data at each site included biomass and grazing animal density observations that describe productivity and grazing intensity at each site. Field samples were collected in the summer of 2014 and 2015 in a nested design, where each site contained five distance classes arrayed at fixed distances from a grazing hotspot (for example, a well, a herder summer or winter camp), and each distance class contained five replicate plots. At each plot location, plant biomass was clipped at the ground from a small 50 cm × 50 cm area, and animal feces were counted on a larger area (10 m × 10 m). Animal feces were attributed to sheep, goats, horses, cows, and camels before being standardized to a single standard livestock unit. For full field sampling methods, see [33].

At the 15 sites, we used RPM to quantify outcomes of twelve fixed-level contrasts, or scenarios of uniform change across all sites, corresponding to plausible future changes in temperature, rainfall, and grazing animal numbers (Table 1). The twelve contrasts are a full factorial design including two levels of temperature (baseline and elevated), two levels of precipitation (baseline and elevated), and three levels of animal density (baseline, elevated, and zero). We derived plausible temperature and precipitation perturbations according to expected future conditions for Mongolia for the time period 2061–2080, a time period which allows us to place our results in context with other recent analyses of climate change in drylands [7,34]. We summarized future climate projections for Mongolia from nine global climate models (GCM). Because our goal was to characterize the outer bound of expected conditions for Mongolia under a reasonable "business as usual" emissions and land-use scenario, we took temperature and precipitation perturbations as the maximum absolute change (for temperature) and maximum percent change (for precipitation) across the nine GCMs for the SSP3-7.0 emissions and land-use scenario. We found that the CanESM5 model projected the largest change in temperature and precipitation for this time period and scenario, showing an increase of 7 °C in average annual temperature and an increase of 18% in annual precipitation (Appendix B, Table A2). We therefore calculated plausible

future climate contrasts from baseline inputs by adding 7 °C to the baseline temperature data, and adding 18% of the baseline value to precipitation inputs (Table 1).

**Table 1.** Fixed-level contrasts explored with the Rangeland Production Model (RPM) at 15 sites spanning a rainfall gradient.

| Contrast | Temperature | Precipitation | Animal Density |
|---|---|---|---|
| A: baseline | Worldclim v.2.0 current conditions | Worldclim v.2.0 current conditions | Total number per site estimated from field records |
| B: elevated temperature | Baseline + 7 °C | Baseline | Baseline |
| C: elevated temperature and precipitation | Baseline + 7 °C | Baseline + 18% | Baseline |
| D: elevated temperature and animal density | Baseline + 7 °C | Baseline | Baseline × 2 |
| E: elevated temperature, precipitation, and animal density | Baseline + 7 °C | Baseline + 18% | Baseline × 2 |
| F: elevated precipitation | Baseline | Baseline + 18% | Baseline |
| G: elevated precipitation and animal density | Baseline | Baseline + 18% | Baseline × 2 |
| H: elevated animal density | Baseline | Baseline | Baseline × 2 |
| I: no grazing | Baseline | Baseline | Zero |
| J: elevated temperature and no grazing | Baseline + 7 °C | Baseline | Zero |
| K: elevated temperature, precipitation, and no grazing | Baseline + 7 °C | Baseline + 18% | Zero |
| L: elevated precipitation and no grazing | Baseline | Baseline + 18% | Zero |

Inputs for the baseline run describe current or recent historical conditions. For climate, baseline conditions describe historical average monthly precipitation over the time period 1970–2000 [32]. For animal density, the baseline density of grazing animals at each site was calculated from empirical dung counts taken at the site during field sampling in 2014 and 2015. The climate and animal density inputs, therefore, pertain to slightly different time periods. Despite the discrepancy, we chose these data sources because for grazing intensity, they represent the best available data, and because the recent historical precipitation data are directly comparable to future climate conditions that were used to drive future scenarios.

We characterized baseline grazing intensity at each site from dung counts recorded during field sampling at the distance class 1500 m from the grazing hotspot. The dung counts roughly followed the site-level rainfall gradient (Appendix A, Figure A1). While yearly livestock census data are available for regional administrative units, the smallest administrative unit for which livestock census data are available (the *soum*) is still a very large area (up to 1 million ha). The dung counts, therefore, provide a much more precise estimate of grazing intensity near the sampling site. However, it was necessary to convert the surveyed dung counts to the animal density input required by RPM; for this, we used soum-level livestock statistics for the year 2015 [35]. We first multiplied the dung counts by a uniform conversion rate, estimated as the mean ratio of total livestock density in the soum containing the site to total dung density at the site. Then, because each site was near a grazing hotspot, and therefore had higher animal density than the administrative unit overall, we multiplied the converted densities by 1.5. This factor, while not based in a quantitative measure, accounts for our understanding that animal density near the grazing hotspot is higher than the average density in the surrounding region. After conversion from dung counts to estimated animal density, animal density at each site ranged from 0.28 to 2.35 animals/ha (Appendix A, Figure A1).

Following initialization of RPM (described below), we ran each contrast under perturbed climate and/or animal density conditions for 24 months. The outcomes that we used to compare the contrasts to baseline were rangeland condition and animal condition.

As an indicator of rangeland condition, we used simulated residual biomass, or biomass left standing after offtake by grazing animals. For animal condition, we used the simplified metric of diet sufficiency. This metric varies between 0 and 1 and describes the extent to which the diet selected by simulated animals meets the animals' energetic needs for maintenance, where both the energetic content of the diet and the energetic demands of maintenance are calculated by the RPM animal physiology sub-model [31]. While energetic requirements for maintenance are static and are determined by the input characteristics of the grazing animal, energetic content of the diet varies at each timestep according to forage availability and protein content. For full sub-model descriptions and equations, see model documentation, Supplementary Materials.

We summarized yearly average diet sufficiency and standing biomass from each model run as the average value across pixels in each site and across months in the second year of modeled conditions. We compared these summary outputs for each contrast to the baseline in terms of percent difference from the baseline (i.e., (yearly average$_{contrast}$—yearly average$_{baseline}$)/yearly average$_{baseline}$* 100).

The plant production and soil nutrient cycling sub-models of RPM reproduce the corresponding routines implemented in the Century model, extended to a gridded format. RPM and Century, therefore, share many input parameters and output values (for full model description and comparison of RPM to Century, see Supplementary Materials.) We leveraged this similarity between the two models to perform model calibration, and also to derive initial conditions for RPM. We derived key parameters controlling vegetation production in RPM by calibrating the same parameters with the Century model over a long spin-up period. We then compared peak biomass estimated by Century to mean empirical biomass per site using total empirical biomass. We first used biomass data collected in 2014 to calibrate parameters related to atmospheric nitrogen deposition (Appendix C, Table A3), adjusting these parameters manually until RMSE between simulated peak biomass and empirical biomass was minimized at 112.2 kg/ha (Appendix C, Figure A7). We chose to focus our calibration on parameters related to atmospheric nitrogen deposition because previous applications of the Century model in Mongolia and nearby regions required modification of these parameter values [36,37], and because biomass production in Century is highly sensitive to these parameters [38]. We then adjusted parameters related to temperature controls on production (Appendix C, Table A3) through comparison to NDVI time series in the study area, to ensure that general seasonal patterns of green-up and senescence were reflected by modeled biomass (Appendix C, Figure A8). For full calibration methods and results, see Appendix C.

Following calibration, we validated RPM under baseline conditions against empirical biomass collected at each site in 2014 and 2015 by driving the model with coterminous, time-varying precipitation inputs from CHIRPS [39]. The model matched empirical biomass well in both years (2014: $\rho = 0.82$, $p < 0.001$; 2015: $\rho = 0.78$, $p = 0.001$). Mean bias across sites was $-43.7$ kg/ha in 2014 (9.4% of mean biomass), and 49.6 kg/ha in 2015 (17% of mean biomass). For full validation methods and results, see Appendix C.

As Century is a point-based model and requires far fewer computational resources than RPM to run, we also used it to generate initial conditions for RPM with a 3300-year long "spin-up" simulation at each site. The spin-up period is necessary to initialize RPM state variables near equilibrium, given historical climate and management conditions. Precipitation inputs for the spin-up were taken from Worldclim v.2.0 [32] in the approximately 1 km$^2$ pixel containing the site centroid. The historical management conditions used in the spin-up included very light year-round grazing (2% of biomass removed each month). All other inputs to the Century model were identical to RPM inputs, and are included in the Supplementary Materials.

*2.4. Gridded Simulation under Future Climate Conditions*

To extend the fixed-level contrasts at the 15 sites to a contiguous, spatially heterogeneous area, we used RPM to project change in biomass and animal diet sufficiency under a

spatially explicit future climate scenario in a 64 million hectare region covering the eastern portion of the country (grey hashed area in Figure 1). We chose this region, the Gobi-Steppe ecoregion, because it is an area of critical concern for herding and wildlife and because it includes the range of annual precipitation seen in the country. We drew input data and parameters for the gridded simulations from the same sources described above for the 15 field sampling sites. We also used the same procedure as described above to generate initial conditions for the simulation, running spin-up simulations with the Century model at regularly spaced points throughout the study area. Following initialization from spin-up simulations under current climate conditions, we ran RPM for 24 months under current and predicted future climate conditions. RPM supports analysis at any spatial resolution, but here we drove the simulation with inputs at a resolution of 10 arc minutes, or 0.167 degrees, to balance computation time with visualization of the study area heterogeneity.

We estimated the change in biomass and diet sufficiency under future climatic conditions with simulations driven by current and future climate data. We used future precipitation and temperature data from Worldclim 2.1, describing monthly average conditions during the time period 2061–2080 under the SSP3-7.0 land use and emissions scenario [32]. Among the nine global climate models (GCM) for which downscaled global data are currently available, we chose the CanESM5 GCM because for the selected time period and scenario, this model shows the largest change in temperature and precipitation for the study area (Appendix B, Table A2). While the modeling procedure that we follow here could be applied to examine the impacts of any future climate scenario, our choice of the SSP3-7.0 scenario and CanESM5 GCM was guided by our desire to explore Mongolia's likely future. The results, therefore, indicate a reasonable outer bound for likely climatic changes under a "business as usual" emissions and land-use scenario.

## 3. Results

### 3.1. Temperature Dominates Precipitation and Grazing Intensity Impacts

Across sites, biomass and animal diet sufficiency were most sensitive to changes in temperature, while the change in biomass and diet sufficiency under increased precipitation and animal numbers was less pronounced (Figure 2). Elevated temperatures led to up to 92% increase in standing biomass (mean across sites: 34%), far exceeding the change in biomass under elevated precipitation (maximum: 30%, mean: 22%) and the increase in biomass under removal of grazing of (maximum: 35%, mean: 15%). All sites showed a decrease in biomass with doubled animal density when this driver was applied alone (maximum decrease: −10%, mean: −3%).

There was large variation between sites in the magnitude of the impact of these fixed-level contrasts, with the strength of response related to annual baseline rainfall at the site (Figure 2; see also Appendix A, Figure A2). There was a strong relationship between site rainfall and response to changes in grazing intensity, where wetter sites showed a larger negative response to increased animal numbers and a larger positive response to removal of grazing. Low-rainfall sites tended to exhibit a larger positive response to elevated temperatures, and there was a similar but less pronounced trend among sites regarding their response to increased precipitation (Appendix A, Figure A2).

When climate and animal density changes were applied two-at-a-time, the resulting shifts in biomass and diet sufficiency were not additive: the combined impact of changes in two drivers tended to be less than the sum of the impacts of each driver applied alone (for full scenario plots, see Appendix A, Figure A3). While elevated temperature or precipitation alone led to a maximum increase in biomass across sites of 92% and 30%, respectively, the maximum increase under their combined effect was 99% (mean increase across sites: 47%). Similarly, while removal of grazing led to a maximum increase in biomass of 35% when applied alone, the maximum increase in biomass under increased temperature and zero grazing was 110% (mean: 51%), and maximum increase in biomass under increased precipitation and zero grazing was 57% (mean: 49%). When elevated temperature or precipitation was paired with increased animal density, sites differed in their response

according to the rainfall gradient: drier sites saw increased production despite higher grazing intensity, while the negative impact of increased grazing intensity dominated at the higher-rainfall sites (Appendix A, Figure A3).

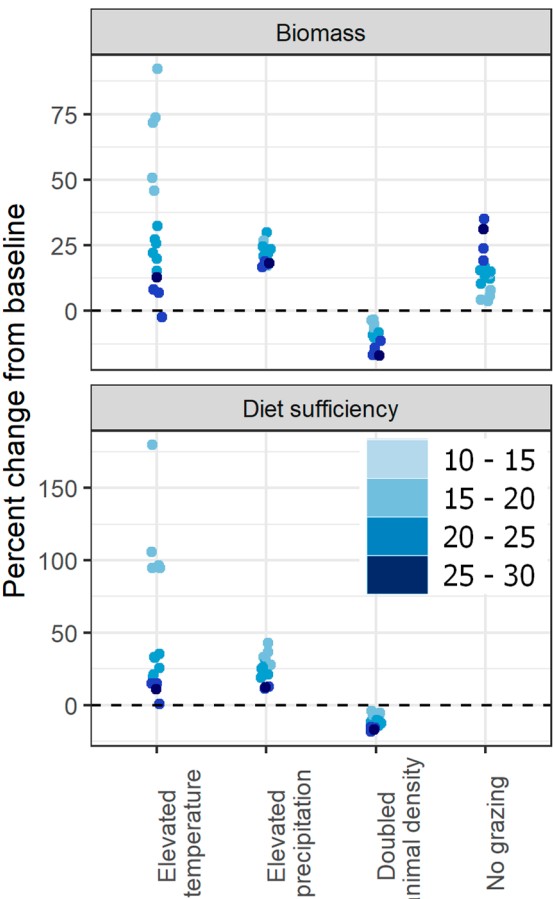

**Figure 2.** Percent change in standing biomass and animal diet sufficiency at 15 sites under fixed-level contrasts of altered temperature, precipitation, and animal density, applied one at a time. Points are colored by average annual baseline precipitation at the site (cm), as shown for the region in Figure 1.

With combined changes in all three drivers, the stimulative impact of elevated temperature and elevated precipitation outweighed the negative impact of increased grazing at the majority of sites (Figure 3). While sites tended to respond similarly to the doubling of animal densities or removal of grazing alone, showing a narrow range of response to changes in grazing as a one-way driver (pink and green areas in Figure 3), the range of responses to changes in grazing when paired with elevated temperature and precipitation was much higher (blue and purple areas in Figure 3). The three-way interactions of these fixed-level contrasts once again showed a strong mediating impact of baseline rainfall on the site's response: low-rainfall sites saw the largest increase in biomass and diet sufficiency (Appendix A, Figure A4).

Biomass and animal diet sufficiency tended to shift in the same direction in response to changes in all factors, although the magnitude of change was often larger for animal diet sufficiency than for standing biomass (Figure 4). This magnified impact on diet sufficiency was especially apparent at low-rainfall sites: at these sites, the amount of change in animal diet sufficiency was up to 2.8 times higher than the corresponding change in biomass. For full results including response of biomass and diet sufficiency to one-way, two-way, and three-way contrasts, see Appendix A, Figures A2–A4.

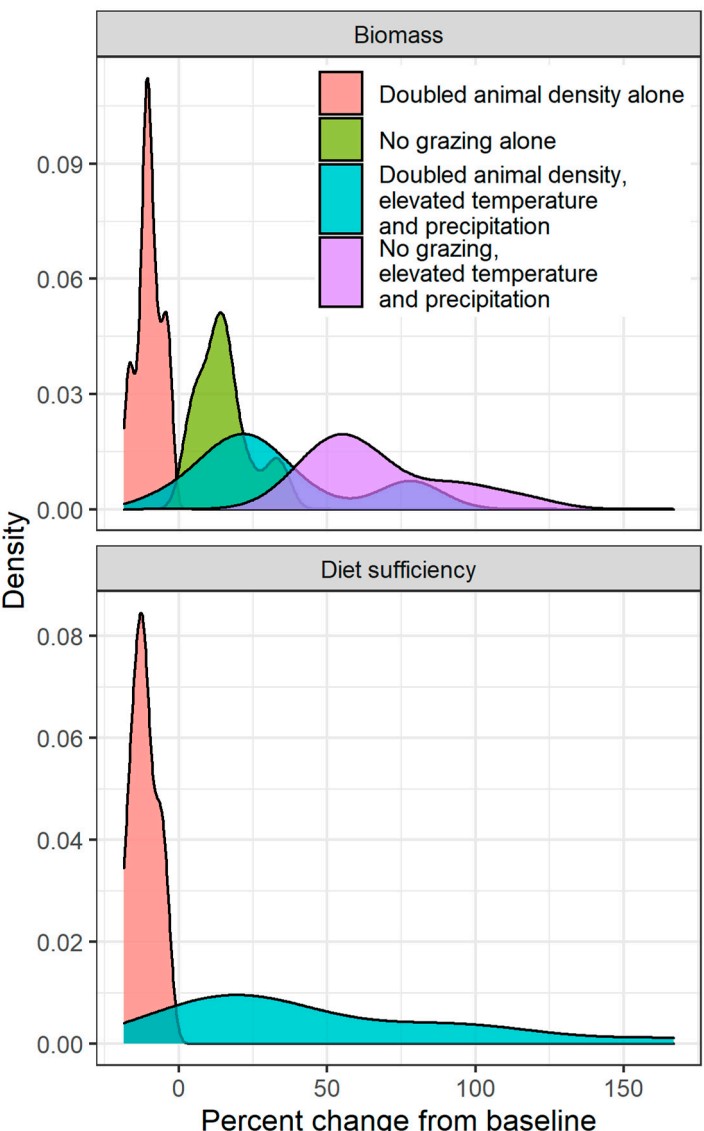

**Figure 3.** Percent change in standing biomass and animal diet sufficiency at 15 sites under fixed-level contrasts of altered grazing intensity, paired with elevated temperature and precipitation or applied alone.

*3.2. Regional Rangeland and Animal Condition under Future Climate Change*

Consistent with fixed-level contrasts, gridded regional conditions under conditions of future climate change showed increased biomass and animal diet sufficiency across most of the Gobi-Steppe ecoregion (Figure 5). While the northern portions of the ecoregion receive the most rainfall under current conditions and saw large increases in productivity under the future scenario (see Appendix A, Figure A6 for baseline values and absolute value change), the greatest percent change in average standing biomass across months of the year was found in the Gobi desert, in the southern part of the study area (Figure 5). In this region, historically the driest portion of the study area, percent change in mean standing biomass was extremely variable across space and ranged from a 73% decrease to 640% increase. This extremely high increase occurred in pixels where the projected change in precipitation and temperature was larger than the mean change for the region explored in the fixed-level contrasts (up to +9.7 °C and 44% increase in precipitation).

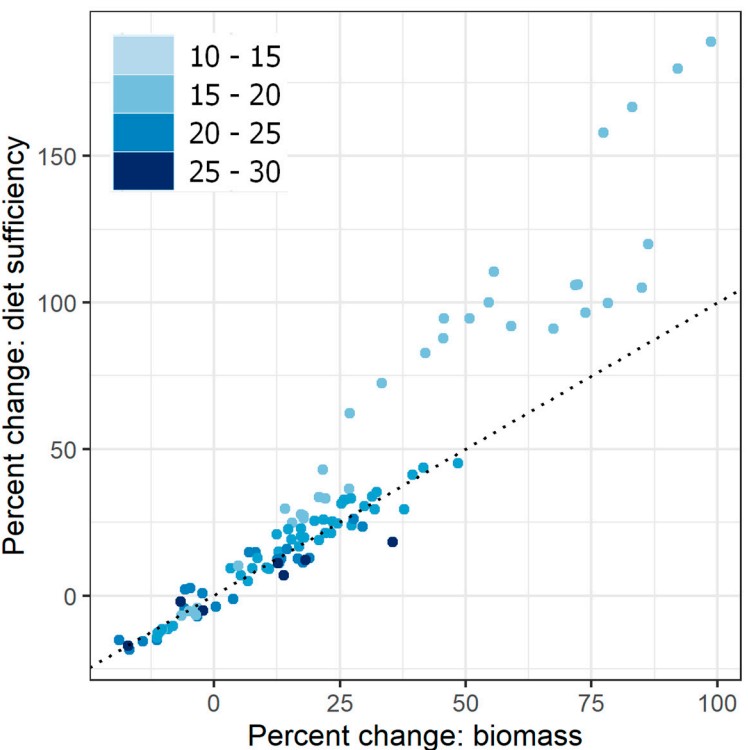

**Figure 4.** Percent change in diet sufficiency and in biomass, for all fixed-level contrasts that included grazing. Points are colored by annual baseline precipitation at the site (cm), as shown for the region in Figure 1. Dotted line indicates 1:1 line.

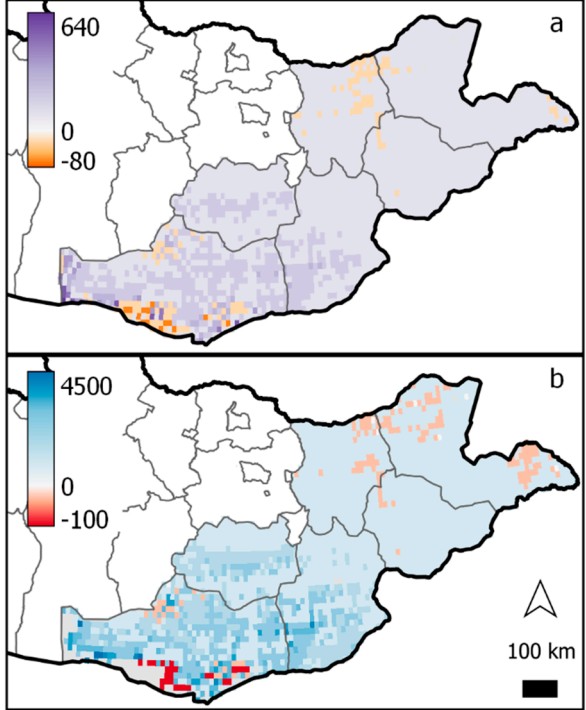

**Figure 5.** Percent change in mean standing biomass (panel (**a**), top) and mean animal diet sufficiency (panel (**b**), bottom) when driving RPM with future climate conditions and current livestock densities, versus current conditions. Negative values indicate a decrease in biomass or diet sufficiency under future climate conditions; positive values indicate an increase.

The spatial pattern of change in animal diet sufficiency was also highly variable and reached its maximum in the Gobi desert, where the percent change in mean diet sufficiency ranged up to 4500% (Figure 5b). In this way, the gridded simulation demonstrated a similar response to the low-precipitation sites in the fixed-level contrasts (Figure 4): in these highly arid locations, percent change in animal diet sufficiency under elevated temperature and precipitation was proportionally greater than percent change in biomass for the same scenario.

## 4. Discussion

### 4.1. Interacting Effects of Climate and Grazing

Our simulations at sites arrayed along a rainfall gradient demonstrated that among the variables tested, temperature was the dominant driver of forage biomass and animal diet sufficiency, followed by precipitation and finally grazing intensity. In addition to the contrasts between these drivers as main effects, RPM showed that site rainfall strongly mediated the impact of livestock on standing biomass. Further, our gridded simulation under predicted future climate conditions suggested that, while predicted future changes in biomass and diet sufficiency will be predominantly positive across a large portion of Mongolia's rangelands, interactions among spatially varying drivers can result in strong spatial heterogeneity.

It is not surprising that increased precipitation in the future would lead to higher productivity: semi-arid rangelands such as those in Mongolia are strongly limited by rainfall [40]. Expectations regarding the impact of elevated temperatures on biomass production are less well established. Conventionally, it is expected that higher temperatures lead to higher evapotranspiration, leading to depressed biomass growth [34]. Controlled warming experiments have demonstrated, however, that temperature as an isolated effect tends to increase aboveground productivity [41]. Our model results support this finding: although RPM did display increased evapotranspiration under elevated temperature, its effects were outweighed by the direct positive effect of increased temperature on biomass growth.

Recent empirical studies have shown that, at the ecosystem level, there is a tipping point temperature at which photosynthesis ceases to respond positively to increases in temperature and begins to slow instead [42]. RPM simulates temperature controls on growth using a similar conceptual model, where temperatures higher than the optimum temperature for growth (here calibrated to 15 °C; Appendix B, Table A2) act to suppress growth. Current average temperatures in Mongolia are well below this tipping point (Appendix A, Table A1), but it is important to note that the stimulative effect of increased temperature on growth detected here is limited by physiological capacity.

Our simulations indicated that, overall, increased grazing pressure had a strongly negative impact on standing biomass, and this intuitive result is supported by observational studies [43]. However, the relative impact of grazing pressure on biomass differed strongly across the precipitation gradient, with more pronounced effects at higher-rainfall sites. This result corroborates statistical analyses of the relative impacts of grazing on rangeland biomass and composition [44]. Our fixed-level contrasts, combining increases in temperature, precipitation and animal density, further illustrated this important interaction: drier sites had increased biomass production despite higher grazing intensity, while the negative impact of increased grazing intensity dominated at the higher-rainfall sites. This was due to the fact that the increase in productivity attributed to changes in temperature and precipitation was much higher than the expected decrease in productivity under increased livestock density. While multi-factor experiments addressing combined impacts of climate change and grazing intensity are rare [14], our modeling approach allows for the flexible exploration of these complex interactions.

RPM showed that diet sufficiency of grazing animals under baseline conditions was strongly related to site precipitation, but all sites had average monthly diet sufficiency values of less than 1; that is, the simulated animals did not meet their forage intake needs,

on average, across months, even at the highest-rainfall sites (Appendix A, Figure A5). While it is possible that uncertainty in model inputs led us to underestimate the nutritional value of forage (see "Modeling limitations and future research directions" below), this finding also supports an important prediction of rangeland nonequilibrium theory: that to be resilient in a highly variable environment, grazing animals must move in response to fluctuating resource conditions driven by abiotic factors [19].

Lacking any spatial information describing fine-scale movement patterns, our gridded simulations assumed that animal density was distributed uniformly across pixels inside grazing areas (see model documentation, Supplementary Materials). In reality, herders in Mongolia move in rotational use between and within seasonal pastures to access spatially and temporally heterogeneous rangeland resources [45]. These seasonal movements allow plants and plant communities to regrow and help the herd to gain the necessary fat and energy to survive the winter months. It is this rotational grazing practice that allows herders to successfully raise animals in Mongolia's very harsh environment, where animals may lose 14–30% of their fall live weight over the winter months [46]. While our analysis did not explicitly demonstrate that a flexible distribution of grazing pressure would allow animals to meet their nutritional needs, our results support the inverse conclusion: when grazing pressure is static over space and time, resources are insufficient to support the animals. Instead, sustainable livestock production in semi-arid rangelands subject to high temporal variability requires flexible and adaptive grazing management.

### 4.2. Modeling Limitations and Further Research Directions

RPM's ability to model the impact of forage availability on predicted animal offtake allows for a more nuanced estimation of the impacts of grazing than a static approach (e.g., [5,47]), where each animal is assumed to consume a fixed amount of forage. However, the additional complexity introduces some important uncertainty. The limitation of forage availability on animals' ability to consume forage is a product of the RPM diet selection sub-model, where forage availability strongly controls intake (see model documentation, Supplementary Materials). The importance of forage availability as a predictor of animal offtake is supported by experimental studies [48], but the modeled relationship is highly sensitive to the standard reference weight of the animal and to forage nutritive content [29]. We validated forage offtake predictions from a previous version of RPM against published feeding trials and verified that when forage availability, crude protein content and digestibility, and animal characteristics such as size and age are precisely known, the model matches empirical values with high accuracy [31]. However, in most applications of RPM, including this application in Mongolia, those inputs are highly uncertain.

Our finding that animals were unable, in most cases, to meet their nutritional needs is apparently contradicted by the situation in reality, where herders successfully raise animals in this environment. While we believe that this result illustrates the importance of livestock mobility, as discussed above, this discrepancy could be further explained by two limitations of our modeling approach. First, it is possible that our assumed level of forage protein content (14.7%) was too low in key periods, failing to capture natural heterogeneity in space and time. Mean protein content of palatable forage in Mongolia ranges from 6.5% to 21.9% depending on forage species, location and season [49–51]. In RPM, both the forage intake and diet sufficiency of grazing animals are highly sensitive to forage protein content (see model documentation; [31]). Second, our modeling approach includes herbaceous vegetation only. Although RPM is adapted from the Century model, which does include woody vegetation, the animal diet selection and diet sufficiency sub-model of RPM limits the model to herbaceous vegetation only. Especially in the driest regions of Mongolia, shrubs are important food for livestock [52], and this modeling limitation means that we were unable to represent this potentially substantial source of nutrition.

In addition to not representing woody vegetation, RPM does not simulate changes in vegetation composition. Studies from Central Asia have suggested that a xerophytization of plant community composition, or an increase in drought-adapted species in moist areas,

can indicate grazing-induced degradation in steppe ecosystems [53]. Changes in plant community composition at the species or functional group level have been shown to be useful indicators of rangeland condition for Mongolia [49,54,55], and the most comprehensive rangeland monitoring program implemented in Mongolia to date uses community composition as a metric of rangeland condition [56]. Such indicator-based approaches, though powerful, can be challenging to generalize to new sites or apply predictively [57], so we consider them complementary to our relatively simple, biomass-based approach.

Simplification of the physical factors also contributed to model uncertainty. The changes in climate drivers that we used RPM to analyze addressed changes in annual temperature and precipitation only, and did not include climatic variability or extreme events. Increased frequency of extreme events may lower productivity aside from changes in mean climate variables [58]. It is already seen under current conditions that, globally, inter-annual variability in precipitation limits livestock production independent of mean annual precipitation [59]. Indeed, a similar application of this model has shown that precipitation variability limits the viability of livestock production nearly as much as total precipitation [31]. Further work should explore the impacts of precipitation variability combined with a mean increase in Mongolia's rangelands.

Finally, our simulations did not include changes in atmospheric $CO^2$ concentration, which could impact both forage availability, through a fertilization effect [58], and forage nutritive content, where an increased C:N ratio could dilute forage nitrogen content and, hence, its digestibility [60]. Through these two mechanisms, elevated $CO^2$ could impact livestock forage consumption and their diet sufficiency. However, the response of rangelands to $CO^2$ enrichment is varied [61], and its impact has not yet been demonstrated for Mongolia.

### 4.3. Implications for Rangeland Management and Policy in Mongolia

Under current animal numbers and for the specific time horizon described by our chosen inputs, future forage biomass and animal diet sufficiency in Mongolia's grasslands are forecast to be predominantly higher. Even under doubled animal density, we found that increased temperature and precipitation drove higher productivity at the majority of sites where we tested this combination. We can also expect higher animal condition in the future: the magnitude of change in diet sufficiency was greater than the magnitude of change in biomass at drier sites, both in the fixed-level contrasts and for the spatially-explicit climate scenario, suggesting that the impacts of climate change on Mongolia's livestock in the future may be magnified relative to the change in biomass alone. With this increased overall productivity, it is reasonable to expect that herders will continue to increase their herd size, as it is a clear trend that higher-productivity areas in Mongolia experience higher animal densities (Appendix A, Figure A1; [33]). Importantly, our results suggest that, with higher productivity, especially at the driest sites, the relative impact of grazing on rangeland condition will increase. Thus, it is possible that, as climate change enables greater overall productivity, Mongolia's rangelands may paradoxically become more vulnerable to degradation through grazing.

The extreme variability evident in our gridded simulations covering the Gobi-Steppe ecoregion should caution against expectations of increasing productivity. Especially in the highly arid southern region, overall increases in productivity were accompanied by higher spatial heterogeneity. Indeed, while the fixed-level contrasts demonstrated consistent trends in biomass and animal sufficiency in response to fixed changes in climate drivers, the spatial heterogeneity evident under more realistic, spatially varying drivers was extreme, and some areas displayed a large decrease in both metrics. Retrospective studies support our findings that Mongolia's rangelands will not exhibit a uniform response to future climate conditions. While productivity in north-central and northeast Mongolia has decreased markedly in recent decades, biomass production in the Gobi desert in Mongolia's southern region has increased [5]. This variability of response across space may arguably be more important for livestock management in the future than the expected mean increase in productivity. Future livestock management policy must support herders' resiliency to such

variability, using the tools that they have honed over centuries of survival in a dynamic and fluctuating environment: flexibility and mobility [62]. Policy is also required that enables coordination among herders, while allowing them to maintain flexibility, to avoid patch scale degradation resulting from overuse by multiple herds. Here, we believe that RPM or a forecasting tool such as the Gobi Forage Livestock Early Warning System [63] can be helpful, enabling coordinating policymakers to identify high-vulnerability areas in advance.

Future livestock management and policy should also consider temporal variability in addition to spatial heterogeneity. There is robust evidence across existing future climate predictions that the variability of precipitation will increase in the future: variability is predicted to manifest at a large range of timescales, from daily to multi-year observations [64]. The expected increase in precipitation variability will lead to more frequent extreme precipitation events, like floods and droughts. For northern latitudes, including Mongolia, the largest increase in precipitation variability is predicted to occur during the summer months [64]; these are the months when rainfall has the greatest potential to impact forage production, so we can confidently expect increasing inter-annual variability in forage production in the future. If increased mean productivity leads to higher animal numbers, this may mean that animals and herders are more vulnerable to drought.

One of the greatest concerns in regard to increased climate extremes is the future occurrence of dzud. The extreme livestock mortality associated with dzud conditions is a product of multiple factors including climate, animal management practices, and social vulnerability [26]. Although herders in Mongolia distinguish many types of dzud, emphasizing that the extreme mortality event may be the result of a number of different combinations of climatic and animal-related factors, the characteristic that is common to all types of dzud is cold winter temperatures [3]. Here, predictions about the future climate offer some hope. While the variability of precipitation is confidently expected to increase in the future, there is no such evidence for the variability of temperature. Global climate models suggest that year-round variability in temperature is likely to decrease in northern latitudes, including Mongolia [65], and seasonal variability of temperature is also expected to change little [66]. In our analysis of the latest future climate projections, we found that winter temperatures are predicted to increase by a minimum of 2.3 °C (Appendix B). Therefore, while drought is likely to become more frequent, extreme cold winters may be less likely in the future. However, this optimistic expectation of reduced risk of dzud should be accompanied by a note of caution: though livestock may be less vulnerable to mortality via cold temperatures, this also means fewer recovery phases for rangeland vegetation.

Other modeling studies of global rangeland productivity under future climate change have pinpointed conversion to woody vegetation as a key driver of changes in forage availability [12]. While conversion from herbaceous to shrub-dominated vegetation may signal degradation in many rangeland environments, such as in North America [67], as noted above, in Mongolia, some shrubs are an important source of forage. Indeed, changes in vegetation composition may act as both a driver of changing productivity and a response to changes in grazing pressure. The conditions under which we found the highest predicted biomass, with elevated temperature and precipitation and the removal of grazing animals, is a useful illustration of an extreme case but not a realistic management option. With their long evolutionary history of grazing, the removal of herbivores from Mongolia's rangelands is neither a practical nor an ecologically desirable outcome [68–70]. Rather than advocating for removal of grazing, our simulations support the idea that flexible and adaptive grazing management is an appropriate management response to extreme spatial and temporal variability. RPM and the modeling approach we introduce here can help to direct sustainable use in this highly variable environment, by identifying areas that are most vulnerable to degradation under expected climatic conditions and uniform, inflexible animal distribution.

## 5. Conclusions

The condition of Mongolia's rangeland, and the livestock and livelihoods that it supports, are certain to experience major shifts in the near future. As anthropogenic carbon emissions continue to accumulate in the atmosphere and social–economic forces drive changes in livestock numbers and management, the response of this large and heterogeneous area will be complex. The response will include interacting and non-additive effects, including a key pattern illustrated by our results: more productive sites will show higher vulnerability to the impacts of grazing. Here we have shown that a dynamic and spatially explicit modeling approach can help to explore the interactions between climate and grazing drivers and their implications for heterogeneous landscapes. Our findings are important because they allow for a better understanding of how Mongolia's forage resources will be impacted in the future by simultaneous changes in climate and animal management, and because they illustrate that climate-driven productivity mediates the impact of livestock grazing on rangeland biomass and the multiple ecosystem services that rangelands support. In the future, we can expect that Mongolia's high-rainfall areas will be vulnerable to grazing impacts, while drier areas will see increased productivity and high variability. It is critical that future rangeland policy in Mongolia supports herders' ability to move flexibly in response to this highly dynamic ecosystem, and the dynamic RPM we have demonstrated here can help to envision future productivity and unravel possible threats.

**Supplementary Materials:** The following are available online at https://www.mdpi.com/article/10.3390/land10040397/s1, Supplement 1: RPM and Century model inputs, Supplement 2: Rangeland Production Model documentation.

**Author Contributions:** Conceptualization, V.A.K., J.A., C.J., O.A. and R.C.-K.; methodology, V.A.K. and R.C.-K.; software, V.A.K.; formal analysis, V.A.K.; investigation, J.A.; writing—original draft preparation, V.A.K.; writing—review and editing, V.A.K., J.A., C.J., O.A. and R.C.-K.; visualization, V.A.K.; supervision, R.C.-K.; funding acquisition, R.C.-K. All authors have read and agreed to the published version of the manuscript.

**Funding:** This work was funded by NASA, grant number NNX17AG56G.

**Data Availability Statement:** The Rangeland Production Model v.0.2.0 is free, open-source and available for download from GitHub (https://github.com/natcap/rangeland_production/releases/tag/0.2.0; accessed on 22 December 2020). Input data that were used to run the model are included in Supplementary Materials. All field data supporting results in the manuscript are available from the authors upon request.

**Acknowledgments:** We are grateful to Lingling Liu for her assistance with NDVI processing. This modeling was informed and motivated by the insights and perspectives of Helen Crowley, Stuart Antsee, Samdanjigmed Tulganyam, James Hamilton, Enkhtuvshin Shiilegdamba, Kirk Olson, and Onon Bayasgalan.

**Conflicts of Interest:** The authors declare no conflict of interest.

## Appendix A. Supplemental Figures and Tables

**Table A1.** Site information for simulated sites. RPM was run at 15 sites numbered from south to north, where biomass and animal dung were sampled at distance transects from grazing hotspots. A gridded simulation was run in RPM covering the Gobi-Steppe ecoregion, a 64 million ha region covering the eastern portion of Mongolia. See Figure 1 in main text for site locations. Mean annual precipitation and temperature from Worldclim v2.0.

| Site. | Latitude | Longitude | Elevation (m) | Precipitation (cm) | Temperature (°C) | Hotspot Type |
|---|---|---|---|---|---|---|
| 1 | 43.52 | 104.22 | 1875 | 14.2 | 3.7 | Camp |
| 2 | 43.73 | 103.6 | 2160 | 16.7 | −0.5 | Well |
| 3 | 43.97 | 105.14 | 1520 | 11.5 | 3.6 | Well |
| 4 | 44.5 | 105.33 | 1310 | 10.4 | 3.8 | Camp |
| 5 | 44.99 | 105.61 | 1200 | 10.8 | 3.5 | Well |
| 6 | 45.63 | 105.84 | 1415 | 13.3 | 1.5 | Well |
| 7 | 45.91 | 106.29 | 1460 | 17.5 | 0.8 | Stable |
| 8 | 46.19 | 106.45 | 1330 | 17.5 | 0.8 | Stable |
| 9 | 46.36 | 106.54 | 1395 | 16.6 | 0.2 | Well |
| 10 | 46.57 | 106.65 | 1360 | 17.7 | 0.2 | Well |
| 11 | 46.79 | 106.64 | 1390 | 18.9 | −0.2 | Camp |
| 12 | 47.06 | 106.57 | 1425 | 21 | −1.0 | Well |
| 13 | 47.34 | 106.63 | 1615 | 24.9 | −1.6 | Well |
| 14 | 47.6 | 106.82 | 1385 | 24.4 | −1.9 | Camp |
| 15 | 48.15 | 106.71 | 1250 | 28.2 | −1.5 | Camp |
| Gobi-Steppe ecoregion * | 45.7 | 109.7 | 1131 | 18.5 | 2.3 | NA |
| Gobi-Steppe ecoregion [†] | 41.6–50.3 | 99.4–119.9 | 560–2748 | 4.1–48.6 | −5.8–9.5 | NA |

* Average across pixels in the region. [†] Range of values across pixels in the region.

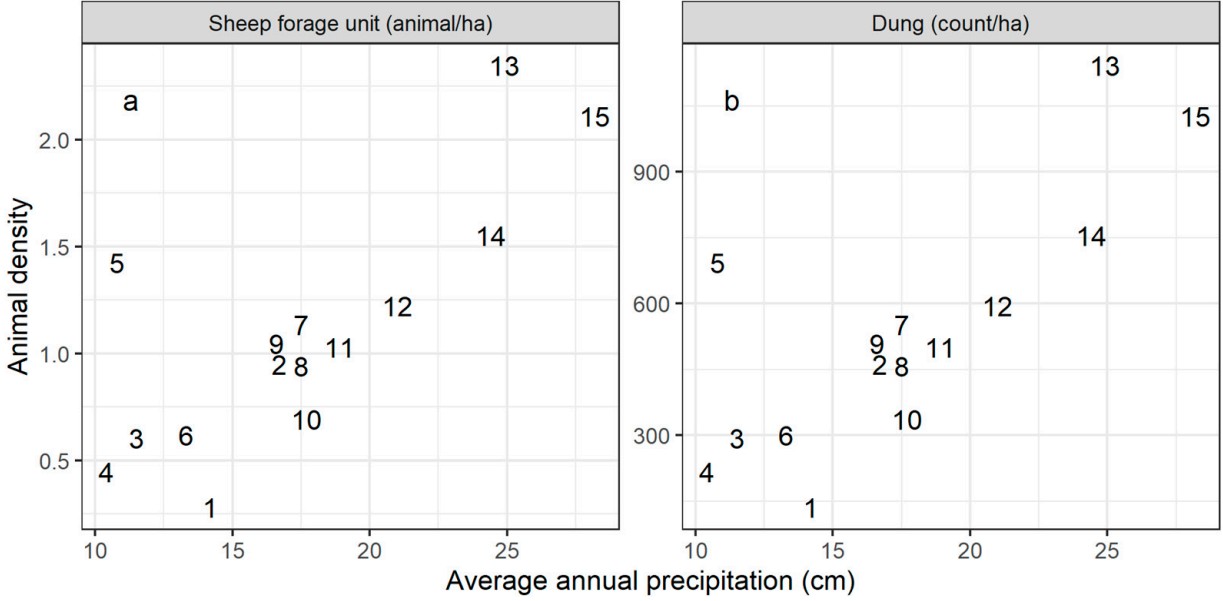

**Figure A1.** Animal density at each site vs. average annual precipitation at the site centroid. Baseline modeled animal density at each site, in sheep forage units per ha (**a**), was estimated from empirical dung counts surveyed at plots 1500 m from a grazing hotspot (**b**). Dung counts were transformed to a standardized livestock unit according to the species that each dung observation was attributed to; for dung count standardization methods, see [33].

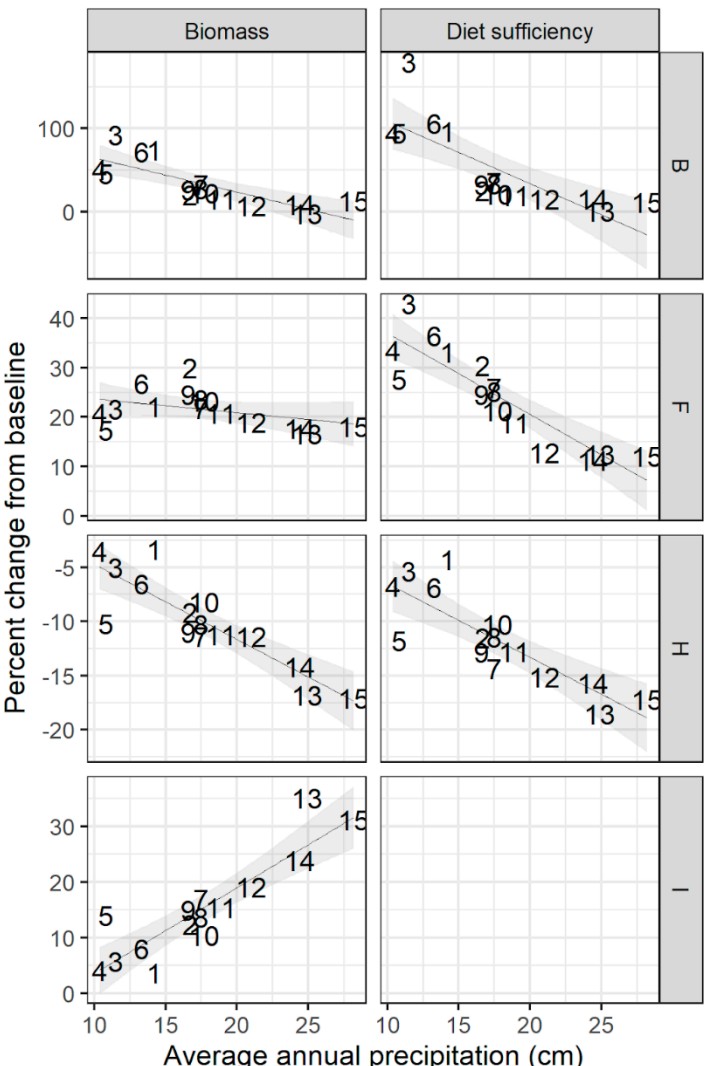

**Figure A2.** Percent change in standing biomass and animal diet sufficiency at 15 sites under fixed contrasts of altered temperature, precipitation, and animal density applied one-at-a-time. Contrast B: elevated temperature; F: elevated precipitation; H: elevated animal density; I: no grazing. Shaded area shows 95% confidence interval around best-fit line. Sites are shown here by their label from south (1) to north (15); see Figure 1 and Table A1 for site locations.

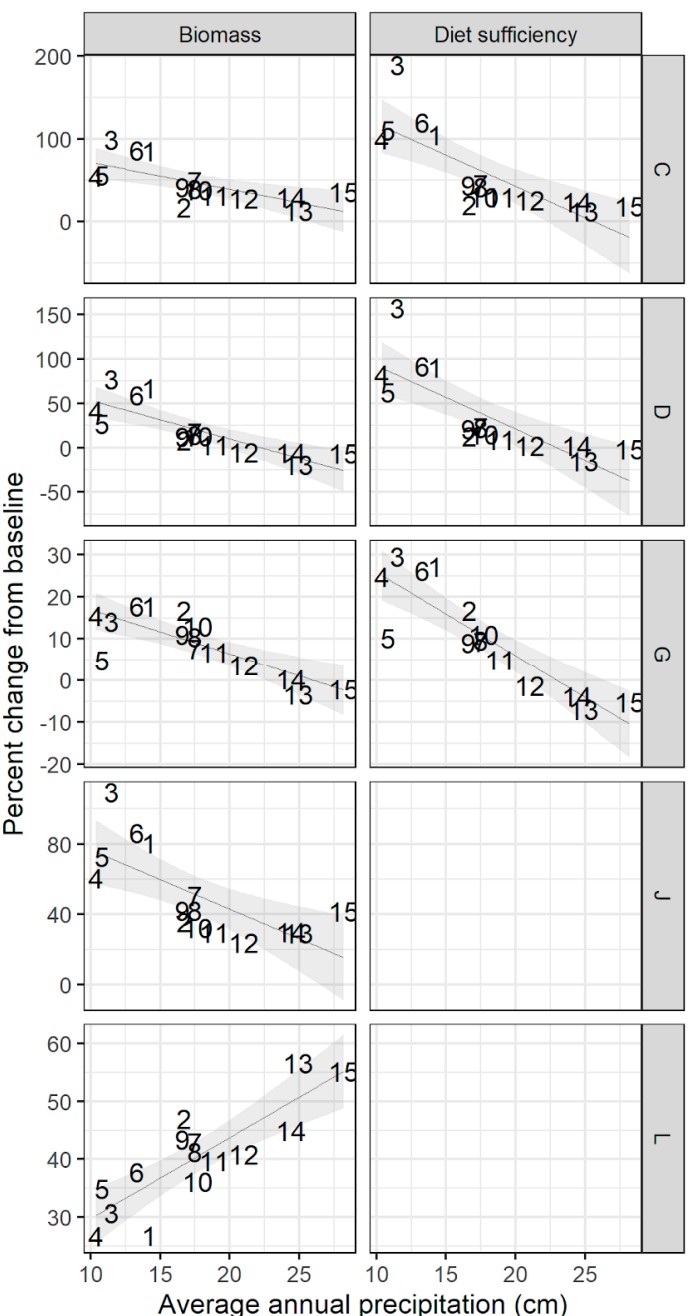

**Figure A3.** Percent change in biomass and diet sufficiency under fixed contrasts that combined changes in climate and animal density factors two-at-a-time. Contrast C: elevated temperature and precipitation; D: elevated temperature and animal density; G: elevated precipitation and animal density; J: elevated temperature and no grazing; L: elevated precipitation and no grazing. Shaded area shows 95% confidence interval around best-fit line. Sites are shown here by their label Figure 1. to north (15); see Figure 1 and Table A1 for site locations.

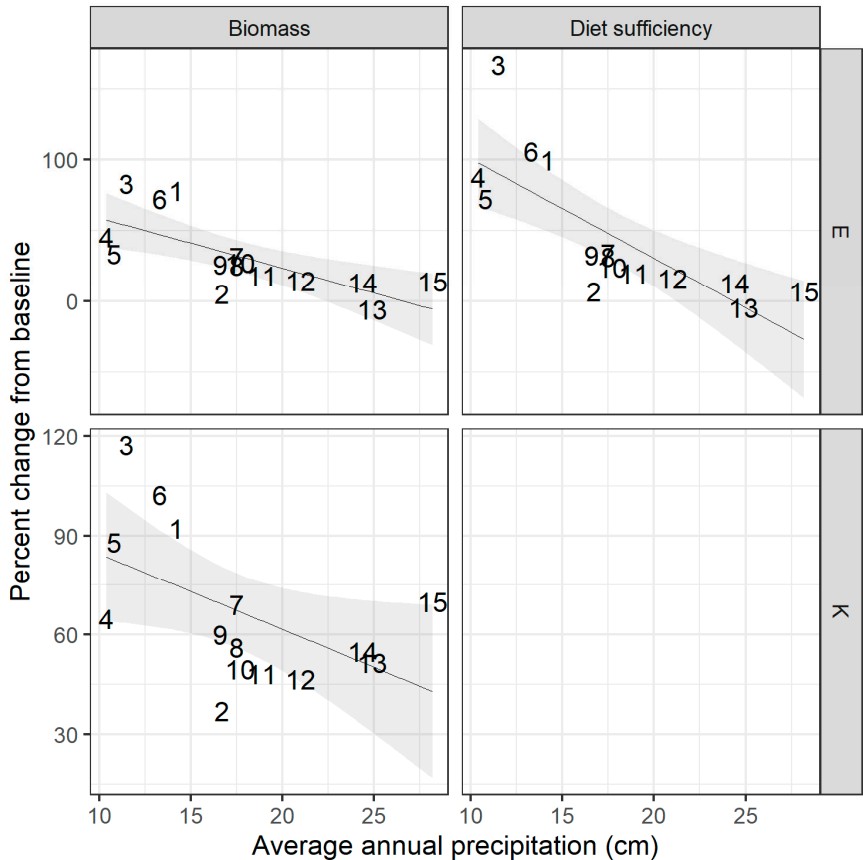

**Figure A4.** Percent change in biomass and diet sufficiency under fixed contrasts of doubled grazing intensity (scenario E) and removal of grazing (scenario K), each paired with elevated temperature and precipitation. Shaded area shows 95% confidence interval around best-fit line. Sites are shown here by their label from south (1) to north (15); see Figure 1 and Table A1 for site locations.

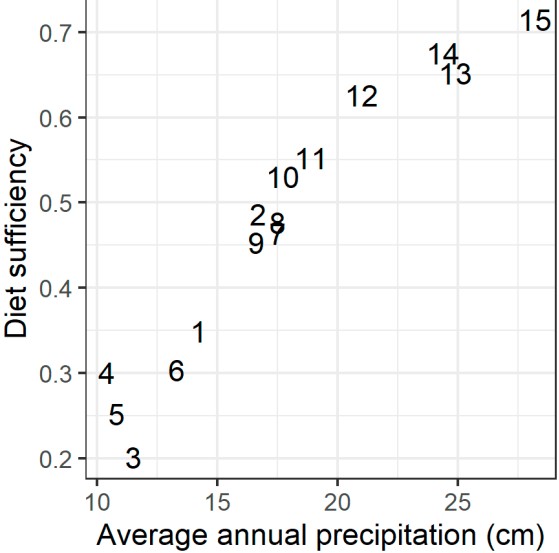

**Figure A5.** Average diet sufficiency at each site in the baseline scenario, under current climate conditions and empirical animal numbers. Sites are arranged here by average annual precipitation. Diet sufficiency values were averaged across twelve months and across pixels inside a 2400 ha simulated area at each site.

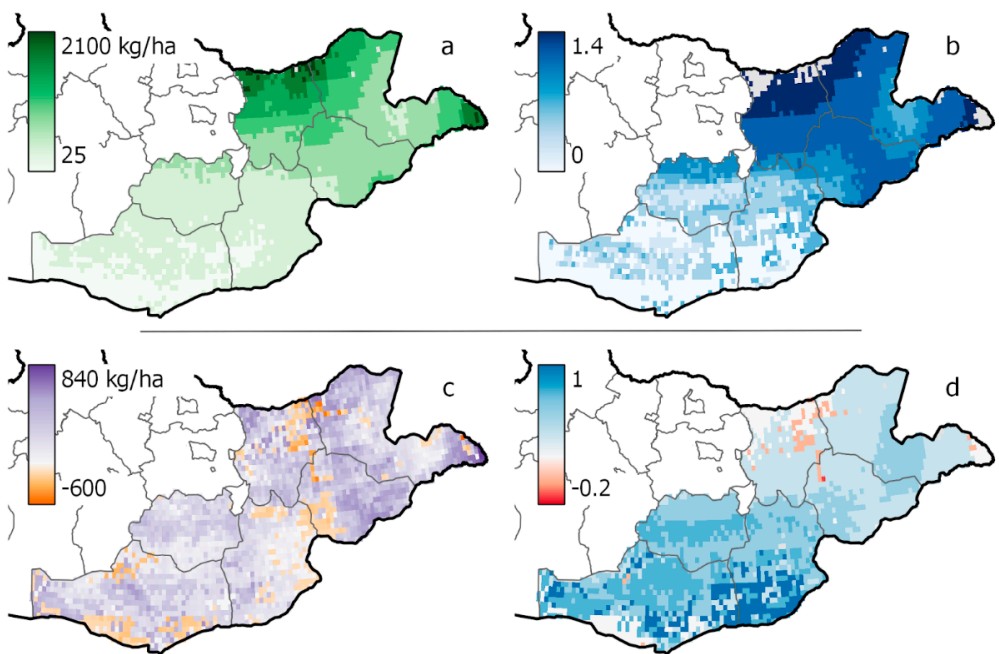

**Figure A6.** Current peak biomass (**a**) and animal diet sufficiency (**b**) for the Gobi-Steppe ecoregion as modeled by RPM when driven by current climate and livestock density. Bottom panels show the absolute value change in peak biomass (**c**) and diet sufficiency (**d**) between RPM simulations driven by future climate conditions as compared to current.

## Appendix B. Synthesis of Future Climate Conditions for Mongolia

We derived realistic bounds for future temperature and precipitation contrasts from outputs of nine global climate models (GCM) from the CMIP6 project [13], downscaled to 2.5 arc minute resolution via calibration to Worldclim v2.1 current climate conditions [32]. We summarized future climate predictions from the nine GCMs for the time period 2061–2080 and the emissions and land-use scenario SSP3-7.0. This scenario ("regional rivalry"), which includes slow economic development, material-intensive consumption, and increasing inequality paired with relatively high carbon emissions, approximates a continuation of business-as-usual for current geopolitical and emissions conditions [71,72].

We analyzed future climate predictions of the nine GCMs for Mongolia for average annual conditions and for winter months only (November through February, following [47]). We first calculated monthly average temperature and precipitation across months and across pixels in Mongolia from current conditions and each of the nine GCMs. We then summarized the change in temperature and precipitation for each GCM relative to current conditions, using absolute values for temperature and relative values for precipitation (following the downscaling methods used to produce the future climate predictions).

On an annual basis, All GCMs predict higher minimum and maximum monthly temperatures for Mongolia (Table A2). Annually, the largest predicted increase is 7.4 degrees for minimum temperature, and 6.3 degrees for maximum temperature. Eight of the nine GCMs predict increased precipitation for Mongolia, up to a maximum of 18%. The CNRM-CM6-1 model alone predicted a small decrease in average monthly precipitation, and the CNRM-ESM2-1 model showed no change in average monthly precipitation (Table A2).

**Table A2.** Change in average monthly temperature and precipitation in Mongolia across all months of the year, relative to current average conditions, according to nine global climate models (GCMs). These refer to future predictions under the SSP3-7.0 scenario and for the time period 2061–2080.

| GCM | Change in Average Monthly Minimum Temperature (°C) | Change in Average Monthly Maximum Temperature (°C) | Percent Change in Average Monthly Precipitation (%) |
|---|---|---|---|
| CNRM-CM6-1 | 3.9 | 3.9 | −1.2 |
| CNRM-ESM2-1 | 4.3 | 4.2 | 0.0 |
| MIROC-ES2L | 4.1 | 4.4 | 0.4 |
| BCC-CSM2-MR | 4.5 | 4.4 | 1.3 |
| MRI-ESM2-0 | 3.5 | 3.7 | 2.8 |
| IPSL-CM6A-LR | 5.3 | 4.8 | 4.6 |
| GFDL-ESM4 | 3.5 | 3.6 | 4.8 |
| MIROC6 | 3.4 | 3.7 | 6.1 |
| CanESM5 | 7.4 | 6.3 | 17.9 |

All models also predict an increase in average temperatures during the winter months (i.e., November to February; [47]). Across GCMs, average winter minimum temperatures are predicted to increase by 3.2 °C (MRI-ESM2-0) to 7.7 °C (CanESM5); average winter maximum temperatures are predicted to increase by 2.3 °C (MRI-ESM2-0) to 5.8 °C (CanESM5).

We generated fixed levels for future climate conditions at each simulated site by taking the maximum change in annual monthly values across models. As RPM requires that maximum monthly temperature must be greater than or equal to minimum monthly temperature, we added a fixed amount of 7 °C to both minimum and maximum temperature inputs.

**Appendix C. Calibration and Validation of RPM and Century**

We calibrated Century and RPM model parameters controlling biomass production and atmospheric nitrogen deposition (Table A3) in two phases. First, we calibrated regional patterns of biomass productivity and its relationship with precipitation by comparing peak biomass modeled by Century to biomass collected at 15 field sites spanning a productivity gradient (Figure 1, main text). Century simulations were driven by monthly climate inputs describing average near-current historical conditions [32] and a historical management schedule that included very low removal of biomass by grazing in all months.

We summarized empirical biomass at each site by calculating mean total biomass across plant functional types, distance from the grazing hotspot, and plot replicates (N = 25, 10 m × 10 m plots per site). We drew empirical biomass from data collected in 2014 only: although vegetation sampling was conducted in 2014 and 2015, the sampling period in 2014 more closely coincided with peak biomass production at each site [44].

After manual parameter adjustment to the values in Table A3, RMSE between mean empirical biomass in 2014 and peak biomass simulated by Century was minimized at 112.2 kg/ha (Figure A7). Mean bias was 26.4 kg/ha, or 5.7% of mean empirical biomass, and the correlation between modeled peak biomass and empirical biomass was strong ($\rho$ = 0.89, $p < 0.001$).

In the second phase of calibration, we calibrated model parameters related to plant production response to temperature through comparison to normalized difference vegetation index (NDVI) time series. NDVI is a remotely sensed index that is widely used to monitor biomass accumulation and vegetation cover in rangelands [73]. We transformed 250 m resolution daily NDVI data tiles from MODIS to remove outliers and fill gaps, following Zhang [74]. We then calculated monthly average NDVI from the fitted and smoothed daily time series, taking the average value across days per month in each pixel, for comparison with monthly model outputs.

**Table A3.** Model parameters that were calibrated with comparison to empirical biomass and NDVI. The same parameter values were used for simulations using Century and RPM.

| Parameter | Definition | Calibrated Value |
|-----------|------------|------------------|
| prdx(1) | Coefficient for calculating potential aboveground monthly production as a function of solar radiation outside the atmosphere ($g/m^2/MJ/m^2$) | 0.2 |
| ppdf(1) | Optimum temperature for growth for parameterization of a Poisson Density Function curve to simulate temperature effect on growth (°C) | 15 |
| ppdf(2) | Maximum temperature for growth for parameterization of a Poisson Density Function curve to simulate temperature effect on growth (°C) | 35 |
| ppdf(3) | Left curve shape for parameterization of a Poisson Density Function curve to simulate temperature effect on growth | 1 |
| ppdf(4) | Right curve shape for parameterization of a Poisson Density Function curve to simulate temperature effect on growth | 2.5 |
| epnfa(1) | Intercept value for determining the effect of annual precipitation on atmospheric N fixation ($g/m^2/yr/cm$ precip) | 0.0001 |
| epnfa(2) | Slope value for determining the effect of annual precipitation on atmospheric N fixation ($g/m^2/yr/cm$ precip) | 0.008 |

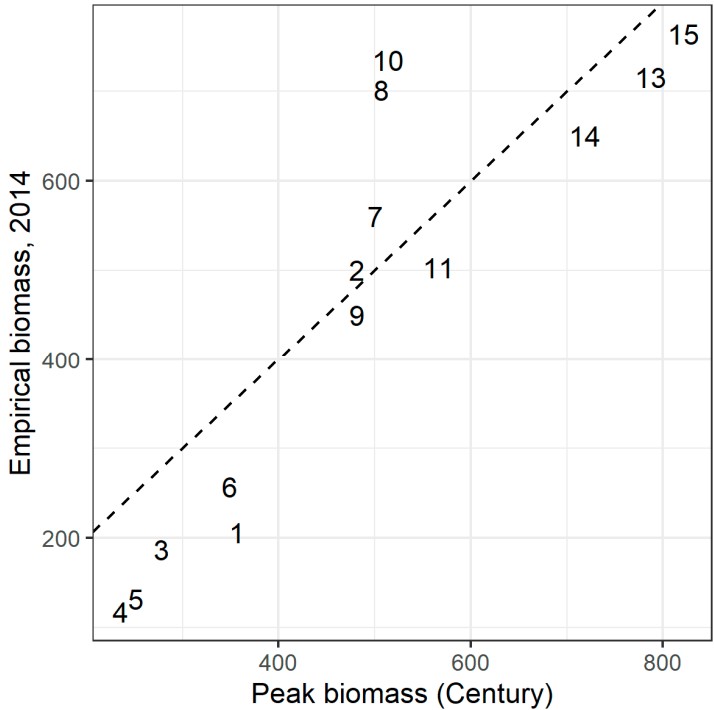

**Figure A7.** Peak annual biomass (kg/ha) at the site centroid predicted by Century, vs. empirical biomass sampled in 2014, after manual calibration. Dashed line shows 1:1 line. RMSE = 112.2 kg/ha.

For this comparison, we used RPM and drove the simulation with time-varying precipitation data from CHIRPS (i.e., not long-term average values; [39]). Simulations began in 2011, following the spin-up period. We included year-round grazing by animals

at uniform empirical density at each site (see main text for derivation of empirical grazing intensity from field data). We compared monthly time series of simulated biomass in the pixel containing each site centroid to NDVI values in the pixel containing the site centroid, for the period covering January 2014 through December 2015. We manually adjusted model parameters related to temperature controls on growth (Table A2) until seasonal biomass production simulated by RPM generally matched the time series of NDVI (Figure A8).

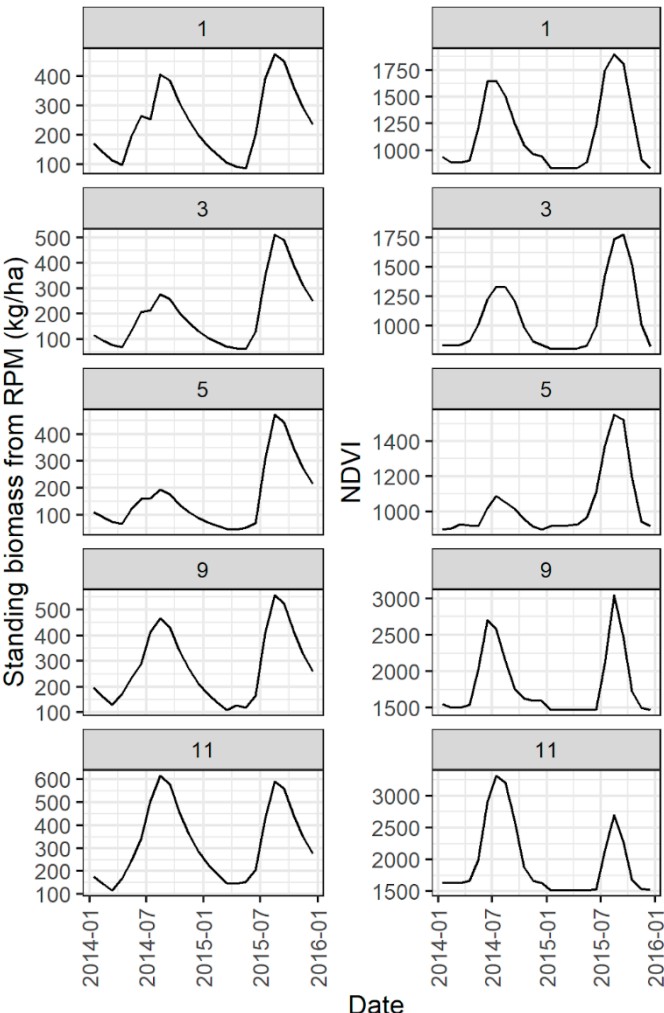

**Figure A8.** Illustration of comparison of standing biomass time series from RPM with NDVI. This comparison was used to calibrate parameters related to temperature controls on production. Time series are drawn from the pixel in the corresponding spatial dataset that contains the site centroid. Although time series at all 15 sites were compared to each other during calibration, only a subset of sites (sites 1, 3, 5, 9, and 11) are shown here for clarity.

*Appendix C.1. Validation of RPM against Field Data*

Following calibration of model parameters, we validated RPM's biomass predictions by comparing simulated biomass at each site centroid to mean empirical biomass per site (Figure A4). For this comparison, we used the RPM simulation driven by time-varying precipitation data from CHIRPS. We began simulations in 2011, following the spin-up period, and compared empirical biomass in both years of field sampling to simulated biomass from RPM in the month that field sampling took place. We included year-round grazing by animals at uniform empirical density at each site (see main text for derivation of empirical grazing intensity from field data).

RPM, when driven by time-varying precipitation and including empirical grazing intensity at each site, matched empirical biomass well in both 2014 and 2015, despite large differences between years (Figure A9). The correlation between simulated and empirical biomass was strongly significant in both years (2014: $\rho = 0.82$, $p < 0.001$; 2015: $\rho = 0.78$, $p = 0.001$). Mean bias across sites was $-43.7$ kg/ha in 2014 and 49.6 kg/ha in 2015, showing that the model did not consistently under- or overestimate biomass in these two years. In general, RPM captured both variability in productivity between sites and inter-annual variability in biomass production within site well. RPM predicted correctly that biomass across sites was lower in 2015 than in 2014, and in general RPM ordered sites correctly, despite large differences in empirical biomass between years.

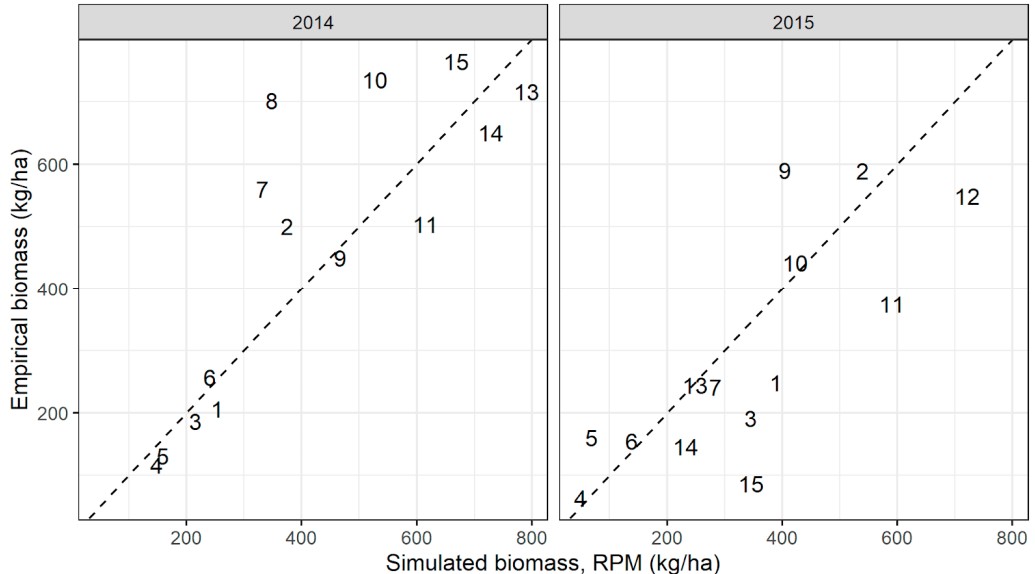

**Figure A9.** Biomass simulated by RPM when driven by time-varying precipitation data from CHIRPS at the site centroid and including empirical animal density at each site, compared to mean empirical biomass within sites.

The performance of RPM when driven by time-varying precipitation and including empirical density of grazing animals at each site was improved over Century, when Century was run with long-term average precipitation and simply included the removal of 3% of standing biomass in each month (Figure A10). While RPM was able to predict biomass in both years with relatively high accuracy, Century's predictions using long-term average precipitation and a simple grazing rule were much less accurate in 2015 than in 2014 (2014: $\rho = 0.82$, $p < 0.001$; 2015: $\rho = 0.59$, $p = 0.03$).

Despite the improved performance of RPM relative to empirical biomass when driven with CHIRPS vs. Worldclim precipitation, we based our scenario analysis on long-term average climate conditions, as described by Worldclim. We made this choice for three reasons: first, because the focus of our study is contrasting rangeland and animal condition over large productivity gradients on an annual average basis, and not on intra-annual variability; second, because the Worldclim current climate dataset includes temperature data that are consistent with Worldclim precipitation data; third, because using Worldclim to characterize current climate conditions allows for a comparison with future climate conditions that are consistent with the Worldclim current dataset.

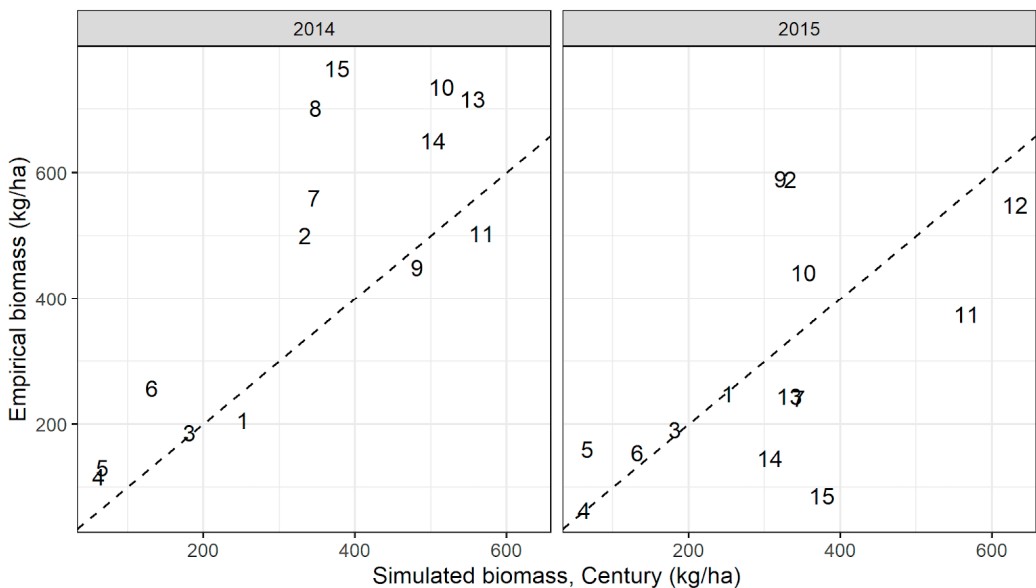

**Figure A10.** Biomass simulated by Century when driven by long-term average precipitation from Worldclim at the site centroid, compared to mean empirical biomass within sites. Simulated biomass shown here was taken from the month when field data were collected at each site (as opposed to Figure A7, which shows peak biomass from Century). Grazing pressure included in Century simulations was year-round removal of 3% of standing biomass.

*Appendix C.2. Validation of RPM against Century*

Nutrient cycling, plant production, and impacts of grazing on plant growth in RPM are reproduced from Century (see RPM documentation for selected Century parameters that were fixed in RPM). We validated RPM's implementation of routines adapted from Century by running each model with identical inputs and starting conditions, and calculating the change in each state variable after one time step. See Tables A1 and A2 in RPM documentation for a list of all state variables shared between RPM and Century. The difference for all state variables between the change predicted by RPM and Century was within the limits of numerical precision ($<0.06$ g/m$^2$).

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
