# Peer review of "Modeling Integrated Impacts of Climate Change and Grazing on Mongolia’s Rangelands"

_land, doi:10.3390/land10040397_

Round 1

Reviewer 1 Report

Through this manuscript, authors explored simultaneous and interacting effects of climate and management changes on Mongolia’s rangeland and livestock production in the future use a gridded, spatially explicit model, the Rangeland Production Model (RPM).

I have just one suggestion: add a study area subsection to Materials and Methods section.

Author Response

We appreciate the reviewer’s comment and we have added a study area subsection (now section 2.2) to the methods section of the manuscript.

Reviewer 2 Report

This is an interesting study that assesses the impact of climate variability and grazing on vegetation biomass and animal diet along a rainfall gradient encompassing 15 sampling sites with varying grazing pressures in Mongolia. The paper is well written and states its aims clearly. However, its methodology lacks some details and flow that adequately describes the model in a way that it is easily understandable and can be replicated. Although a lot of supplementary information is provided to support the model rationale, it is useful to readers to have some basic information on the manuscript without continually looking for it in the supplementary material. Additionally, the connection between the Century model, RPM and integrating of field data is unclear and its discussion throughout the methodology section is confusing to the reader. It would be great if the description of the models and how they are connected can be explained in a clear way. It is also unclear what variables were considered static and which ones were dynamic during the model simulation. Below are specific comments on the paper.

Specific review comments

Line 41 – provide examples of ‘social changes’ that have increased livestock production in Mongolia for those not familiar with Mongolia.

Line 97 – what is the source of the climate and soil data used in the Century model?

Section 2.1 What is the source of the input data for climate, soil, animal protein etc input data to the model

Line 93 – what is the size of the simulation grids?

Line 96 – is the RPM global or local? If the model is global, was calibration of the herbaceous vegetation done to suit the functional types of herbaceous vegetation of the 15 sampling points in this study?

Line 124-specify what existing data was used to calibrate and validate the model. How was the calibration done? How was the validation done?

Lines 127-128 are not clear. Were the five replicates within the 50x50 plot? And what do ‘distance classes’ mean and represent?

Line 150-153 – baseline climate data was collected for 1970-2000 and baseline animal density were done in 2014 and 2015, the difference in time between these data is 15 years. Isn’t this discrepancy in time too major to affect the accuracy of the results?

Line 149-160-the frequency of dung counts is not discussed and the estimation of grazing intensity using dung counts is unclear.

Lines 165-167 – where is the 1.5 coming from? Aren’t there published data that can be used to estimate animal density near grazing hotspots in Mongolia?

Lines 173-176 – provide more information on the differentiation of dead and live biomass in relation to the months. Was this separation in months done when estimating dung counts, and grazing intensity?

Lines 176-181 – estimation of animal condition is not clear. For example, what type of diet indicated a poor animal condition and what diet indicated a healthy animal. Was the animal condition static throughout the simulation or did it vary with the offtake amount? More explanation is needed if the model is to be replicable. What was the source of the estimates for ‘energetic needs for animal maintenance’?

Line 192-195 – what was the purpose of calibrating atmospheric nitrogen with biomass? Why was nitrogen alone used and no other atmospheric parameters?

Author Response

We thank the reviewer for their careful attention to the model description and for the many helpful comments.  We have edited the methods section heavily in response to each specific comment given by the reviewer. We have also edited the manuscript throughout to improve our explanation of the relationship between RPM and Century, and to better describe how the field data were integrated into the modeling workflow.

Specific review comments

Line 41 – provide examples of ‘social changes’ that have increased livestock production in Mongolia for those not familiar with Mongolia.

We have added to the text two salient examples of recent market-related changes in Mongolia: the privatization of livestock production in 1990, and increasing global demand for cashmere.

Line 97 – what is the source of the climate and soil data used in the Century model?

We have edited this section to indicate that all model input sources are described briefly in section 2.3 of the methods, and in full in supplementary material.

Section 2.1 What is the source of the input data for climate, soil, animal protein etc input data to the model

We have added text here to indicate that the source of model inputs are described in section 2.3 of the methods.

Line 93 – what is the size of the simulation grids?

We have added a line of text to clarify that the spatial resolution of RPM is flexible and determined by model inputs.

Line 96 – is the RPM global or local? If the model is global, was calibration of the herbaceous vegetation done to suit the functional types of herbaceous vegetation of the 15 sampling points in this study?

We have added text here to clarify that like the spatial resolution of the model, the spatial extent of RPM simulations are determined by model inputs. Input parameters and calibration were derived relative to the dominant herbaceous vegetation at the 15 sampling sites, and are described in full in supplementary material.

Line 124-specify what existing data was used to calibrate and validate the model. How was the calibration done? How was the validation done?

We have added text here to indicate that calibration and validation results are described briefly at the end of section 2.2, and in full in Appendix C.

Lines 127-128 are not clear. Were the five replicates within the 50x50 plot? And what do ‘distance classes’ mean and represent?

We have edited this section to clarify the nested sampling scheme.

Line 150-153 – baseline climate data was collected for 1970-2000 and baseline animal density were done in 2014 and 2015, the difference in time between these data is 15 years. Isn’t this discrepancy in time too major to affect the accuracy of the results?

This is an excellent point and we have added text to acknowledge this discrepancy. We have also added an explanation to the text as to why we chose to use these two data sources: for livestock density, the 2014-2015 estimates are the best available data; for precipitation, the 1970-2000 data are directly comparable to future climate conditions that we used to drive future scenarios.

Line 149-160-the frequency of dung counts is not discussed and the estimation of grazing intensity using dung counts is unclear.

We have heavily edited the description of field sampling methods, including dung counts, and clarified the methods that we used to convert dung counts to animal density. Figure A1, Appendix A, shows the frequency of dung counts at each site, and we have mentioned this in the text.

Lines 165-167 – where is the 1.5 coming from? Aren’t there published data that can be used to estimate animal density near grazing hotspots in Mongolia?

We have edited this section to include our assertion that the dung counts taken at field sampling sites represent a much more precise estimate of grazing intensity at the site than other available data. We have also edited the section to clarify our reasoning behind the methods we used to convert dung counts to animal density.

Lines 173-176 – provide more information on the differentiation of dead and live biomass in relation to the months. Was this separation in months done when estimating dung counts, and grazing intensity?

We appreciate the reviewer pointing out that this differentiation is not relevant. It is only part of the spin-up procedure that was used to establish initial conditions for the simulations that are the focus of the paper. We have removed this differentiation from this section of the text and clarified that the spin-up procedure included very light year-round grazing.

Lines 176-181 – estimation of animal condition is not clear. For example, what type of diet indicated a poor animal condition and what diet indicated a healthy animal. Was the animal condition static throughout the simulation or did it vary with the offtake amount? More explanation is needed if the model is to be replicable. What was the source of the estimates for ‘energetic needs for animal maintenance’?

We have amended this section to clarify that the energy content of the diet and the energetic requirements of maintenance are calculated by the RPM animal physiology sub-model. We have expanded the description to indicate that energetic requirements of maintenance are static according to animal characteristics, while the energy content of the diet varies at each timestep according to simulated forage availability and protein content.

Line 192-195 – what was the purpose of calibrating atmospheric nitrogen with biomass? Why was nitrogen alone used and no other atmospheric parameters?

We have edited this section to clarify that we chose parameters related to atmospheric nitrogen deposition because total biomass production in Century is highly sensitive to these parameters, and because existing applications of Century in Mongolia and nearby regions required modifications of these parameter values.

Reviewer 3 Report

The authors investigated the impacts of both climate change and grazing on Mongolia’s rangelands. The Rangeland Production Model (RPM) has been applied to integrate impacts of climate change and animal management changes on rangelands and animal conditions. The manuscript topic seems interesting but needs to be carefully revised and improved, particularly the introduction section. I recommend the paper for publication after major and minor revisions described below are addressed.

Line 49- Bring full term for the IPCC at first appearance (Intergovernmental Panel on Climate Change).

Line 51 – It has been mentioned that temperature and precipitation will increase under “high-emissions scenario”. Which specific scenario are you referring to? Is it RCP8.5? It is important to be specific with predictions of climate models and emission scenarios.

Line 55- Please re-write the following sentence as the meaning is not clear:

“Climatic variability in semi-arid and arid rangeland systems is a determining factor both of forage production and of the impacts of grazing intensity on the rangeland ecosystem”.

Line 79 – Be careful when using the word “new”. The RPM has been widely used in other studies.

Lines 79 to 88 – Please highlight the research gap and novelty of your research. What are the main differences between your study and previous conducted research? What is the novelty aspect of your research?

Please improve the introduction section by discussing other research that used climate models to assess climate change impacts on rangelands in Mongolia. What models/method have been applied? Did they only use RPM modelling approach to analyse results?

Line 94 – Replace “Century” by “century”. Please revise the entire manuscript accordingly.

Line 95- Replace “submodel” by “sub-model”. Please revise the entire manuscript accordingly.

Line 103- Add space after “root”

Line 109 – Replace “fifteen” by “15”. Please revise the entire manuscript accordingly (line 123 etc.)

Lines 116 to 121 – Avoid too long figure caption

Lines 132 to 145 – Pleas justify why have you defined future temperature by adding 7 degrees to current condition?

Why have you added 18% to current precipitation to obtain future precipitation? Please justify the selection of added values (7 degrees and 18%).

Lines 225 to 232 – Justify your climate model and emission scenario selections. Why have you selected CanESM5 climate model among nine GCMs? Why have you selected SSP3?

Author Response

We thank the reviewer for their many helpful comments. We have addressed the specific comments below, including a substantial revision and expansion of the introduction section according to the reviewer’s suggestions.

Line 49- Bring full term for the IPCC at first appearance (Intergovernmental Panel on Climate Change).

The correction has been made.

Line 51 – It has been mentioned that temperature and precipitation will increase under “high-emissions scenario”. Which specific scenario are you referring to? Is it RCP8.5? It is important to be specific with predictions of climate models and emission scenarios.

We have revised this sentence to reflect that the cited projections are derived from ensemble-mean projections under the RCP8.5 scenario.

Line 55 - Please re-write the following sentence as the meaning is not clear:

“Climatic variability in semi-arid and arid rangeland systems is a determining factor both of forage production and of the impacts of grazing intensity on the rangeland ecosystem”.

The sentence has been re-written.

Line 79– Be careful when using the word “new”. The RPM has been widely used in other studies.

We have revised this sentence to indicate that this study is the first to use an updated, gridded version of RPM.

Lines 79 to 88 – Please highlight the research gap and novelty of your research. What are the main differences between your study and previous conducted research? What is the novelty aspect of your research?

Please improve the introduction section by discussing other research that used climate models to assess climate change impacts on rangelands in Mongolia. What models/method have been applied? Did they only use RPM modelling approach to analyse results?

We have revised the introduction to include a more complete discussion of existing studies that address climate change and its expected impacts on rangelands in Mongolia. As suggested by the reviewer, we have broadened this section to highlight the research gap and novelty of our research: first, that our study is the first to include analysis of the most recent global climate projections from the CMIP6 project; and second, that our study explicitly addresses the relative and combined impacts of changes in climate and grazing intensity.

Line 94 – Replace “Century” by “century”. Please revise the entire manuscript accordingly.

While we respect the reviewer’s suggestion, we note that the majority of published papers that use the Century model refer to it using the capitalized form. See for example the citations listed below. We prefer to follow this convention.

  • Berardi, Danielle, et al. "21st‐century biogeochemical modeling: Challenges for Century‐based models and where do we go from here?." GCB Bioenergy 12.10 (2020): 774-788.
  • Oelbermann, Maren, et al. "Estimating soil carbon dynamics in intercrop and sole crop agroecosystems using the Century model." Journal of Plant Nutrition and Soil Science 180.2 (2017): 241-251.
  • Bruni, Elisa, et al. "Additional carbon inputs to reach a 4 per 1000 objective in Europe: feasibility and projected impacts of climate change based on Century simulations of long-term arable experiments." Biogeosciences Discussions (2021): 1-35.

Line 95- Replace “submodel” by “sub-model”. Please revise the entire manuscript accordingly.

The correction has been made.

Line 103 - Add space after “root”

The correction has been made.

Line 109 – Replace “fifteen” by “15”. Please revise the entire manuscript accordingly (line 123 etc.)

The correction has been made.

Lines 116 to 121  – Avoid too long figure caption

We have shortened and re-written the figure caption.

Lines 132 to 145 – Pleas justify why have you defined future temperature by adding 7 degrees to current condition?

Why have you added 18% to current precipitation to obtain future precipitation? Please justify the selection of added values (7 degrees and 18%).

We have edited this section to better describe the methods by which we obtained these temperature and precipitation contrast levels. Related to the reviewer’s following suggestion, we have clarified that our goal was to characterize reasonable outer bounds of a “business as usual” scenario for Mongolia.

Lines 225 to 232 – Justify your climate model and emission scenario selections. Why have you selected CanESM5 climate model among nine GCMs? Why have you selected SSP3?

We have clarified in this portion of the text that our choice of the SSP3 scenario and the CanESM5 GCM was guided by a desire to explore reasonable outer bounds of a “business as usual” scenario for Mongolia. However, we also note in the text that a similar modeling procedure could be applied to any future scenario for which climate data exist.

Round 2

Reviewer 2 Report

The revised version of the manuscript titled ‘Modeling integrated impacts of climate change and grazing on Mongolia’s rangelands’ is way better than the previous version. Additional details by the authors have clarified the rationale of the methodology in relation to the study objectives and study area.

Author Response

We thank the reviewer for their comments, which helped to improve the manuscript.

Reviewer 3 Report

Dear editor,

Although authors have addressed most of my comments, my major following comment has not yet been addressed:

" Lines 79 to 88 – Please highlight the research gap and novelty of your research. What are the main differences between your study and previous conducted research? What is the novelty aspect of your research?"

Additionally, the introduction section still requires more improvements as my following comment was not addressed properly:

"Please improve the introduction section by discussing other research that used climate models to assess climate change impacts on rangelands in Mongolia. What models/method have been applied? Did they only use RPM modelling approach to analyse results?"

I recommend the manuscript for publication after addressing the stated above comments.

Author Response

We thank the reviewer for their attention to our manuscript. We have edited the introduction in accordance with the reviewer's comments, as detailed below.

" Lines 79 to 88 – Please highlight the research gap and novelty of your research. What are the main differences between your study and previous conducted research? What is the novelty aspect of your research?"

We draw the reviewer’s attention to lines 69-72 in the revised manuscript, where we state in the context of previous studies that our manuscript is the first to our knowledge to analyze the latest climate projections for Mongolia from the CMIP6 modeling project.

We also draw the reviewer’s attention to lines 73-75 in the revised manuscript, where we clarify that our study addresses a gap in existing research: we address simultaneous changes in climate and grazing pressure.

In the final paragraph of the introduction (lines 102-107), we highlight these two areas where our study fills gaps in the existing research. First, that our study is the first to analyze the latest climate projections for Mongolia from the CMIP6 global climate modeling project. Second, that our study addresses simultaneous changes in climate and animal management in Mongolia in the future and therefore allows for better understanding of important interactions between these two major drivers.

Additionally, the introduction section still requires more improvements as my following comment was not addressed properly:

"Please improve the introduction section by discussing other research that used climate models to assess climate change impacts on rangelands in Mongolia. What models/method have been applied? Did they only use RPM modelling approach to analyse results?"

We draw the reviewer’s attention to lines 57-71 in the revised manuscript, which was added in response to the reviewer’s comment. In this paragraph, we have added a complete review of the previous studies that exist to our knowledge that used climate models to assess climate change impacts on rangelands in Mongolia. This paragraph describes that previous studies used the Century ecosystem model paired with climate projections underlying the Fifth Assessment report (AR5); the G-range global rangeland model paired with climate projections underlying the Fifth Assessment report (AR5); and heuristic understanding of plant growth under climate projections underlying the Fourth Assessment report (AR4).